# Epileptic EEG Signal Detection Using Variational Modal Decomposition and Improved Grey Wolf Algorithm

**DOI:** 10.3390/s23198078

**Published:** 2023-09-25

**Authors:** Yongxin Sun, Xiaojuan Chen

**Affiliations:** 1College of Electronic Information Engineering, Changchun University of Science and Technology, Changchun 130022, China; sunyongxin@bcnu.edu.cn; 2College of Physics and Electronic Information, Baicheng Normal University, Baicheng 137099, China

**Keywords:** epilepsy, improved grey wolf algorithm, VMD

## Abstract

Epilepsy does great harm to the human body, and even threatens human life when it is serious. Therefore, research focused on the diagnosis and treatment of epilepsy holds paramount clinical significance. In this paper, we utilized variational modal decomposition (VMD) and an enhanced grey wolf algorithm to detect epileptic electroencephalogram (EEG) signals. Data were extracted from each patient’s preseizure period and seizure period of 200 s each, with every 2 s as a segment, meaning 100 data points could be obtained for each patient’s health period as well as 100 data points for each patient’s epilepsy period. Variational modal decomposition (VMD) was used to obtain the corresponding intrinsic modal function (VMF) of the data. Then, the differential entropy (DE) and high frequency detection (HFD) of each VMF were extracted as features. The improved grey wolf algorithm is adopted for a selected channel to improve the maximum value of the channel. Finally, the EEG signal samples were classified using a support vector machine (SVM) classifier to achieve the accurate detection of epilepsy EEG signals. Experimental results show that the accuracy, sensitivity and specificity of the proposed method can reach 98.3%, 98.9% and 98.5%, respectively. The proposed algorithm in this paper can be used as an index to detect epileptic seizures and has certain guiding significance for the early diagnosis and effective treatment of epileptic patients.

## 1. Introduction

The risk factors of epilepsy include a family history of epilepsy, craniocerebral injury, febrile convulsion, neonatal disease and pregnancy risk factors [1]. At present, China has carried out in-depth studies on antiepileptic drugs, as well as treatment methods such as additive therapy, minimally invasive surgery and comorbidity prevention and treatment [2]. The disease is characterized by recurrent attacks and no fixed signs before the onset. According to statistics, about 80% of patients with epilepsy will have abnormal EEG during the interictal period [3]. Therefore, EEG monitoring is the main means of judging the morbidity of epilepsy patients. There are many kinds of epilepsy, and EEG is not intuitive [4]. It is inefficient and easy to misjudge epilepsy symptoms by relying on the subjective experience of professional physicians. Therefore, real-time detection and high-precision classification recognition of EEG signals in epilepsy patients is the main direction of current research [5].

For decades, in order to improve the diagnostic performance of EEGs, researchers have extracted effective information from EEG signals from many aspects [6]. The parameters such as peak amplitude, slope and coefficient of variation can be used to characterize the temporal characteristics of EEG signals [7]. The characteristic extraction method based on the spectral value is adopted to extract the corresponding mean value, standard deviation, entropy and other parameters from the power spectrum of each sub band, which can integrate the time-domain feature into the frequency-domain feature [8]. Considering the non-stationary and unilateral characteristics of the EEG signal, time-frequency analysis was carried out [9]. With the development of research, people find that the brain can be regarded as a nonlinear dynamic system, so an entropy-based feature extraction method is introduced [10]. The authors of [11] proposed the concept of approximate entropy from the perspective of measuring the complexity of time series, which is vulnerable to the influence of data length. Subsequently, Richman et al. proposed the concept of sample entropy to remedy the defect of approximate entropy. Some researchers have found that the experimental measurement of the volatile half-life of different brain electrical channels uses the autoregressive moving average generalized autoregressive conditional heteroscedasticity (ARMA-GARCH) model. The confidence interval is constructed using the delta method and asymptotic method to compare the half-life [12].

In order to improve medical efficiency, automatic epilepsy recognition based on EEG has been widely studied [13]. At the same time, in the process of epilepsy treatment, patients with ineffective antiepileptic drug treatment need surgical treatment [14]. Surgical treatment requires accurate identification of surgically excised epileptogenic foci. Epileptogenic foci can be divided into initial foci, irritant foci, epileptogenic lesion area and functional deficiency area. The primary area is the most effective marker and is regarded as a surrogate indicator of epileptogenic foci. Clinically, different types of tracers can be used to determine the location of the initial area at the onset of epilepsy [15]. Therefore, SOD has important clinical significance. Grewal designed a seizure monitoring system that adjusts parameters based on users’ own data [16]. The sensitivity of the detection system is 89.7%, and the average time delay is 17.1 s. Sorensen et al. proposed an SOD method based on the matching pursuit algorithm. The sensitivity of this method can reach 78~100%, and the delay time is controlled within 5~18 s [17]. In recent years, high-sampling rate devices have become increasingly popular, and researchers have found that the high-frequency oscillating rhythm in epileptic EEG above 80 Hz can also serve as a new indicator of the origin region [18].

The core problems of the automatic detection of epilepsy can be summarized into two sub-problems: feature extraction and classifier design [19]. Deep learning integrates feature extraction and classifier design through a neural network. This paper will compare the advantages and disadvantages of various methods in the automatic detection of epilepsy from the perspective of feature extraction and classifier selection [20]. At present, the most effective classification methods are the combination of time-frequency domain and nonlinear dynamics [21]. This method does not take into account the spatial characteristics of signals and fails to achieve high precision classification and recognition rate. The classification process is usually dichotomous or directly tripartite, ignoring the continuity of seizures. Many studies have focused on artifacts in EEG signals, such as eye movement, muscle and baseline artifacts, which pose challenges in their detection and attenuation. For instance, Ghosh et al. [22] proposed a robust method that can automatically detect and remove eyeblink and muscular artifacts from EEG using a k-nearest neighbor (KNN) classifier and a long short-term memory (LSTM) network. Through parameter validation, this method preserves structural correlations, minimizes frequency distortion and optimally removes artifacts from the EEG. Experimental results have demonstrated that the proposed method minimizes distortion caused by eye artifacts to the greatest extent while eliminating both blink-related and muscle artifacts. Mandhouj et al. [23] developed a deep convolutional neural network (CNN) model that can effectively detect and classify epileptic seizures based on EEG spectral images. The experimental results proved to be a powerful tool for EEG signal classification with an average accuracy of 98.22%. Thangavel et al. [24] developed an automated system for detecting epileptic EEG with or without IEDs. These results pave the way towards the automated detection of epilepsy. Ein et al. [25] proposed a computer-aided seizure diagnosis classification system based on feature extraction and channel selection using EEG signals. The results showed that the proposed approach based on the ensemble classifier is better classified than the other classifiers in all metric parameters.

In this paper, variational modal decomposition and the improved grey wolf algorithm were used to detect epileptic EEG signals. Aiming at the problem of epileptic EEG signals, an improved detection mode is proposed. Firstly, the VMD method is used to extract the features. The improved grey wolf algorithm is used for channel selection [26]. Experimental detection uses a support vector machine (SVM) as the classifier. The experiment validates the 10-fold cross-validation. This algorithm is compared with other algorithms to optimize the detection method of epileptic EEG signal, which has very important clinical significance for the study of epileptic diagnosis and treatment.

This paper consists of five main parts: the first part is the introduction, the second part is the state of the art, the third part is the methodology, the fourth part is the result analysis and discussion and the fifth part is the conclusion.

## 2. State of the Art

### 2.1. Automatic Detection Process of Epilepsy

Automated detection of epilepsy is a key component of epilepsy diagnostic and therapeutic research. It involves several complex steps designed to accurately identify epileptic EEG signals. The following is a general summary of the key components involved in the process.

(1) Signal Acquisition: Signal acquisition serves as the initial stage in epilepsy detection. EEG abnormalities during epileptic seizures manifest as specific waveforms and patterns. The collected EEG data can be used as input for the automatic detection of epilepsy. (2) Signal Preprocessing: The preprocessing of EEG signals is a crucial step in this process. EEG signals, in their original form, are non-stationary and dynamic, characterized by low amplitudes. Therefore, in order to effectively analyze EEG signals, it is necessary to perform preprocessing such as artifact removal while retaining relevant information. (3) Feature Extraction and Selection: Following signal preprocessing, the next step is feature extraction. Feature extraction aims to comprehensively characterize EEG signal patterns, highlight distinctions between epileptic and normal states, and effectively discriminate epileptic seizures. (4) Classification Model Learning and Evaluation: The final stage involves the use of classification models to learn and evaluate EEG signals for epilepsy detection. These models fall into two main categories: statistical analysis and machine learning.

#### 2.1.1. Data Collection and Input

EEG abnormalities in epileptic seizure states are mainly manifested as spike wave, slow spike wave or various rhythms. Spike wave is one of the typical characteristics of epileptic discharge in EEG signal. It is manifested by negative surface deflection caused by cortical surface orientation, steep ascending and descending branches, and the overall shape of thorns like spikes. It has a wave change in the EEG signal caused by abnormal firing of nerve cells in the cortex. The spike wave period is usually 20~70 ms, which is prominent in the background signal. The amplitude is more than 20 μV, which is more than 1.5 times that of the background signal. Epileptic spines and spikes in epileptic seizures are usually extracted clinically. The pathological information related to epilepsy was obtained by analyzing epileptic spines and spikes.

EEG signals can be divided into intracranial EEG and scalp EEG. Intracranial EEG is to record electrical activity signals of different parts of the brain by placing strip or mesh electrodes in the epidural area through cranial drilling or craniotomy. Scalp EEG, on the other hand, uses EEG caps to place electrodes at fixed locations on the human scalp and record electrical activity. EEG signals collected by these methods can be used as input signals for the automatic detection of epilepsy.

In the current study, part of the data are from the hospital’s non-public patient data, and most of the data are from public datasets. Table 1 lists the datasets commonly used in studies related to the automatic recognition of epilepsy.

#### 2.1.2. Data Preprocessing

The original EEG signal is non-stationary and dynamic. Scalp EEG itself has a small amplitude, and the acquisition program will be affected by 50 Hz or 60 Hz power frequency signal. In addition, seizures are accompanied by loss of consciousness and body movements, and the acquisition process can easily experience interference due to uncontrollable factors, so the scalp EEG is often random to a certain extent. In order to study and analyze the characteristics of EEG signals in the future. Firstly, it is necessary to remove the artifacts from the original EEG signal without losing the effective information, so as to improve the classification accuracy. The main approach involves three steps. The characteristic values of EEG signals, such as amplitude, area under equivalent curve, etc., are considered abnormal when the signal exceeds a certain threshold. Common artifacts include ocular artifact, myoelectric artifact, electrode shift, emission artifact and diffuse fast sharp alpha wave. Experts’ experience can be used to screen the typical interference characteristics so as to remove the corresponding artifacts. Most of the effective information of EEG is concentrated between 1 Hz and 60 Hz. Therefore, band-pass filtering is a common method for EEG signal preprocessing. The method based on the Kalman filter can enhance the signal-to-noise ratio (SNR) of the EEG signal and improve the detection rate of the peak wave. In addition, half-wave processing is a common EEG signal processing method used to smooth out spikes and burrs in the EEG signal.

#### 2.1.3. Feature Extraction and Selection

It is an important step to realize the automatic detection of epilepsy to extract effective features as a classification basis by analyzing the signals. Reasonable and typical epileptic EEG features can comprehensively characterize EEG signal patterns. At the same time, it can effectively describe the difference in EEG signals between epileptic seizure and normal state, highlight the difference between spine wave and background signal and help the classification model to discriminate epileptic seizure effectively. The quality of the feature seriously affects the final classification performance.

In general, there are four categories of features used to detect epilepsy: the time domain characteristics of sequence waveform and sequence cross-correlation; the frequency domain feature represented by power spectrum density characterizing signal energy; time-frequency domain features of original EEG signals converted by time-frequency transform method; and the nonlinear characteristics of signal uncertainty measurement such as sample entropy, permutation entropy, Hurst parameter and higher-order spectrum analysis based on nonlinear analysis.

The feature matrix can be extracted by singular value decomposition, principal component analysis, independent component analysis and correlation analysis to reduce the feature dimension. In feature selection, one-way analysis of variance (ANOVA) is the most commonly used method.

#### 2.1.4. Classification Model Learning and Evaluation

Classification models can be divided into two types: statistical analysis and machine learning. For the statistical analysis model, in addition to directly specifying the threshold value of signal characteristics to judge the attack state, it also includes a distribution test and correlation analysis. Morphological analysis, time-frequency analysis and similar techniques compared the differences in characteristics between the detected signal and the template epileptic seizure signal. This enabled automatic epilepsy detection.

With the development of artificial intelligence, machine learning models are becoming widely used in the automatic detection of epilepsy, including traditional machine learning that categorizes manually extracted features directly, deep learning based on a neural network, transfer learning to overcome individual differences in EEG, multi-view learning integrating multiple feature views, integrated learning with multiple base classifiers and active learning with optimized annotation samples.

## 3. Methodology

### 3.1. Dataset

The dataset used in this paper is the CHB-MIT dataset, which is a collection of EEG signals from Boston Children’s Hospital included in the MIT EEG database. The dataset is composed of scalp EEG recordings from children with refractory epilepsy. A total of 23 recordings from 22 subjects were included. All EEGs were acquired using 10–20 international standard electrode positions, and EEGs were recorded using 18/23 leads with a sampling frequency and resolution of 256 Hz and 16 bits, respectively.

### 3.2. Feature Extraction

#### 3.2.1. Variational Mode Decomposition (VMD)

The VMD method is an adaptive and completely non-recursive modal variational and signal processing method. The VMD method is well suited for feature extraction from EEG signals due to its adaptability in determining the number of modal decompositions, its ability to overcome band aliasing issues and its suitability for positive definite problems, which are important considerations when working with EEG data. These advantages make it a valuable tool in our research for the accurate detection of epileptic EEG signals.

First, regarding the structural variational problem, suppose the original signal f is decomposed into z components. The decomposition sequence is a finite bandwidth modal component with a central frequency. Find the sum of the estimated bandwidths of each mode to minimize it. The constraint condition is that the sum of all modes is equal to the original signal, and the corresponding expression is Equation (1).
(1)$$\underset{\left.\left\{p_{z}\right\}\cdot |\theta_{z}\right\}}{min}\;\{\sum_{z}\;\;{∥\partial_{n}\left[{\left(\delta_{n}+j/\pi n\right)}^{*}p_{z}(n)\right]e^{-je_{z}n}∥}_{2}^{2}\}\;s.t.\;\sum_{z=1}^{z}\;\;p_{z}=f$$
where f is the original signal and z is the signal component. The decomposition sequence is a finite bandwidth modal component with a central frequency. To solve Equation (1), the Lagrange multiplication operator *λ* is introduced to transform the constrained variational problem into an unconstrained variational problem, and the augmented Lagrange expression in Equation (2) is obtained.
(2)$$L\left(\left\{p_{z}\right\},\left\{\omega_{z}\right\},\lambda \right)=\alpha \sum_{z}\;{∥\left[\left(\partial_{,}+\frac{j}{\pi n}\right)*u_{z}(n)\right]e^{-j\omega_{z}n}∥}^{2}+{∥f(n)-\sum_{z}u_{z}(n)∥}_{2}^{2}+\left(\lambda (n),f(n)-\sum_{z}u_{z}(n)\right)$$
where α is the quadratic penalty factor, which is used to reduce the interference of Gaussian noise. λ represents the change in the extreme value of the objective function when the constraints change. When $\omega_{z}$ increases or decreases by one unit value, f changes λ accordingly.

The alternate direction multiplier iterative algorithm combined with Parseval and Fourier isometric transform was used to optimize the modal components and central frequencies, and the saddle points of the augmented Lagrange function were searched. The expressions of $p_{z}$, $\omega_{z}$ and λ after alternate optimization iteration are Equations (3)–(5), respectively.
(3)$${\hat{p}}_{z}^{t+1}(\omega )\leftarrow \frac{\hat{f}(\omega )-\sum_{i=z}\;{\hat{u}}_{i}(\omega )+\hat{\lambda }(\omega )/2}{1+2\alpha {\left(\omega -\omega_{z}\right)}^{2}}$$
(4)$$\omega_{z}^{t+1}\leftarrow \frac{\int_{0}^{\infty }\;\omega {\left|{\hat{u}}_{z}^{t+1}(\omega )\right|}^{2}d\omega }{\int_{0}^{\infty }\;{\left|{\hat{u}}_{z}^{t+1}(\omega )\right|}^{2}d\omega }$$
(5)$${\hat{\lambda }}^{t+1}(\omega )\leftarrow {\hat{\lambda }}^{t}(\omega )+\gamma \left(\hat{f}(\omega )-\sum_{z}\;{\hat{u}}_{z}^{t+1}(\omega )\right)$$
where γ is noise tolerance, which meets the fidelity requirements of signal decomposition. The Fourier transform is performed on ${\hat{p}}_{z}^{t+1}\;(n)$, ${\hat{p}}_{i}\;(n)$, $\hat{f}\;(n)$ and $\hat{\lambda }\;(n),\;respectively$.

The VMD method overcomes the phenomenon of band aliasing because it minimizes component bandwidth. The principle based on a single channel solution is superior to the ICA method, which is only suitable for positive definite problems. In conclusion, the VMD method can meet the requirements of removing EEG epilepsy in terms of the frequency band separation effect and application conditions.

Generally, sample entropy, approximate entropy, fuzzy entropy and energy entropy are often used as the characteristics of EEG signals. However, because the energy distribution of the EEG signal components in epileptic patients is different from that of the normal EEG signal components, the energy entropy method is used in this paper to solve this problem. The energy entropy is calculated according to Equation (6).

The EEG signal f(n) was decomposed by VMD.
(6)$$f(n)=\sum_{z=1}^{z}\;\;p_{z}(n)$$
where $p_{z}(n)$ is the *n* VMF component.

The energy of the VMF component is shown in Equation (7).
(7)$$E_{z}=\sum_{n}\;{\left|p_{z}(n)\right|}^{2}$$

The proportion of the VMF energy spectrum to the total energy spectrum is shown in Equation (8).
(8)$$U_{z}=E_{z}/\sum_{z=1}^{Z}\;E_{z}$$

The energy entropy of each VMF component.
(9)$$B_{z}=-U_{z}{log}_{10}U_{z}$$

According to the definition of energy entropy, the more the energy of each frequency band accounts for the total energy, the smaller the energy entropy will be. Because the VMD method has a good frequency band separation function, it can separate the epilepsy component from the EEG signal well.

The detailed process of the VMD-VMF method is shown in Figure 1. First, the EEG signals containing epilepsy were filtered. Then, the power frequency interference was removed and the useful EEG frequency band signals were extracted. On the basis of the comprehensive consideration of computational complexity and algorithm performance, the parameter K = 6 of VMD was set to carry out a six-layer variational mode decomposition, and the variable mode function (VMF) component was obtained.

#### 3.2.2. Features

In this paper, the differential entropy (DE) and Higuchi fractal dimension (HFD) of each VMF are extracted to form the feature set. Then, the calculation process of these two features will be introduced, respectively.

(1)The differential entropy (DE)

The DE feature of the EEG data was calculated using the short-time Fourier transform (STFT) and a non-overlapping Hamming window (1 s) and averaged across five bands. The differential entropy equation is Equation (10).
(10)$$b(I)=-\int_{S}\;f(i)logf(i)di$$

To simplify the calculation, the differential entropy can be simplified as Equation (11), assuming that the EEG signals follow a Gaussian distribution of I~T (μ, σ_2_).
(11)$$b(I)=\frac{1}{2}ln(2\pi e\sigma^{2})$$

According to Equations (10) and (11), differential entropy characteristics can be obtained. 

(2)The Higuchi fractal dimension (HFD)

HFD is an algorithm used to calculate the fractal dimension of time series. A time series I (1), I (2)... is set with length N, I (T). A matrix $I_{z}^{w}$ is obtained by using the delay method to reconstruct the time series, and its form is Equation (12).
(12)$$I_{z}^{w}:I(w),I(w+z),I(w+2z),\cdots ,I\left(w+int\left(\frac{T-w}{z}\right)z\right),w=1,2,\cdots ,z$$

The curve length $L_{w}\;(z)$ of each $I_{z}^{w}$ can be calculated using the following equation to obtain Equation (13).
(13)$$L_{w}(z)=\frac{1}{z}\left[\left(\sum_{x=1}^{int\left(\frac{T-w}{z}\right)}\;\;|I(w+xz)-I(w+(x-1)z)|\right)\times \frac{T-1}{xtn\left(\frac{T-w}{z}\right)z}\right]$$

The total sequence curve length can be approximated by the average length of *z* delayed generation sequence curves $L(z)=\frac{1}{z}\sum_{w=1}^{z}L_{w}(z)$.

For different values of *z*, a set of curve data about *z* and *L(z)* is obtained, and L *L*_h_
*(L(z))~L*_h_
*(1/z)* curves are then drawn. If it is a straight line, *L(z)* has the following relationship with *K*: $L(z)\sim z^{-FD}$.

The fractal dimension of time series can be obtained by the linear fitting of data to get *L*_h_
*(L (z)) = FD* × *L*_h_
*(1/z) + C*.

The algorithm has high accuracy in calculating the fractal dimension of time series, but the parameter *z_max_* is not specified in the implementation process of the algorithm. In the application of the algorithm, several authors use the method of guessing and trying to determine the value of *z_max_*.

### 3.3. Classification

In this paper, we use a support vector machine (SVM) to classify EEG signals, which is a supervised generalized linear classifier. It has now been progressively theorized and has become part of the theoretical foundation of statistics. The main idea of it is based on the structural risk minimization principle. It creates an optimal hyperplane by solving a convex quadratic programming problem so that the intervening edge between positive and negative examples is maximized. In a practical classification problem, for a given training dataset $D=\left(i_{1},j_{1}\right),\left(i_{2},j_{2}\right),\ldots ,\left(i_{T},j_{T}\right),{\;j}_{x}\in \{0,\;1\}$, the hyperplane is denoted as
(14)$$m^{⊤}i+h=0$$
where m is the normal vector and h is the displacement term. The distance of any point i in the sample space to the hyperplane (m, h) is denoted as
(15)$$r=\frac{\left|m^{T}+h\right|}{∥m∥}$$

Then, the problem of finding the maximum separated hyperplane is summarized as solving the constrained optimization problem:(16)$$\underset{m,h}{min}\;\frac{1}{2}∥m{∥}^{2}$$
(17)$$s.t.\;j_{x}\left(m^{T}i_{x}+h\right)\geq 1,x=1,2,\cdots ,w$$

In this paper, $i_{x}$ in the training dataset represents the EEG signal features extracted from the *x*th person, and $j_{x}$ is the sample label. If $j_{x}$ = 0, it represents a healthy person; if $j_{x}$ = 1, it represents an epileptic patient.

In order to accurately classify the EEG signals of epileptic patients, two types of samples are set up here: one is a normal EEG sample from a conscious healthy person; the second is the EEG samples of epileptic patients at the onset of epilepsy. The EEG samples are classified using SVM as a classifier. The accurate detection of epileptic EEG signals was achieved, and the classification graph is shown in Figure 2.

### 3.4. EEG Channel Selection

According to the traditional method, all channel information of the EEG is directly input into the system classifier. The disadvantage of this is the large amount of data, which will directly affect the efficiency of the SVM classifier. Therefore, the EEG channels are filtered and those containing valid features are retained. 

The grey wolf optimization (GWO) algorithm is usually used to solve continuous optimization problems. In this paper, the GWO algorithm is used for channel selection. The position vector carried by each wolf represents a channel, which is an alternative solution. In general, the search range of the continuous algorithm is arbitrary. However, the solution vector of the channel selection problem should be a numeric vector containing only 0 s and 1 s. Therefore, in the binarization algorithm, the search range of each dimension of the alternative solution is restricted to the interval [0, 1]. In order to binarize the grey wolf position, it must be converted to a binary vector for position updating. The calculation procedure is Equation (18).
(18)$$I^{d}(n+1)=\left\{\begin{array}{l} 1,\;S\left(\frac{I_{1}^{d}+I_{2}^{d}+I_{3}^{d}}{3}\right)>r_{3}\\ 0,\;other \end{array}\right.$$
where r_3_ is the random vector in the range [0, 1] and d is the dimension of the search space. Equation (19) is the calculation of the S function.
(19)$$S(x)=\frac{1}{1+e^{-10(i-0.5)}}$$

The binarization method of GWO is called the binary grey wolf optimization. First, the wolves are randomly initialized (0 or 1). Secondly, the classification accuracy of the KNN classifier [22] is used as the fitness value to evaluate the merits and demerits of the eigenvalue. The KNN algorithm is a widely used classification technique in machine learning. It operates on the straightforward principle of considering the similarity between data points to make predictions. The KNN algorithm employs a two-step approach for classification. In the first step of the KNN algorithm, it searches for the nearest data points in the feature space to the data point that needs to be classified. In the second step, once the K nearest neighbors are identified, the algorithm assigns the class label to the data point in question based on a majority voting scheme. This majority voting mechanism makes KNN a straightforward and effective classification method. According to the calculated fitness value, the three wolves with the highest fitness value were selected as the leader. For each wolf, *I*_1_, *I*_2_ and *I*_3_ were calculated. The grey wolf position is updated by Equation (18). Then, the fitness values of grey wolves were calculated, and the positions of α, β and δ were updated. The algorithm is repeated until the termination condition is satisfied. In order to improve the efficiency of the algorithm while guaranteeing the classification accuracy, five channels are finally selected as the input of the SVM classifier in this paper.

## 4. Result Analysis and Discussion

The data in this paper were extracted from each patient’s preseizure period and seizure period of 200 s each, with every 2 s as a segment, meaning 100 data points of each patient’s health period and 100 data points of each patient’s epilepsy period could be obtained. Variational modal decomposition and the improved grey wolf algorithm were used to classify epileptic EEG signals. In processing the EEG, 5 of the 23 channels were finally selected as inputs to the SVM classifier by the grey wolf algorithm. In order to make the algorithm performance fully verified, 10-fold cross-validation was used in the experiments.

In the seizure detection of epilepsy, EEG examination is of great value to the judgment of its etiology and seizure type. The most common is scalp electroencephalography, which places electrodes on the scalp to record changes in the electrical potential of brain cells as they move.

In this paper, the public EEG data are used for the CHB-MIT dataset. First, VMD decomposition is performed on the raw EEG data to obtain VMFs. Then, extraction features are performed, i.e., the differential entropy (DE) and Higuchi fractal dimension (HFD) of each VMF are extracted as features. The channels of the EEG signals are selected by the grey wolf optimization algorithm. Finally, the EEG signals are classified by the SVM classifier. The overall flowchart of the algorithm in this paper is shown in Figure 3. The results of the VMD decomposition of the original EEG signal are shown in Figure 4. It can be seen from Figure 4 that the signals of different frequencies contain different features.

Figure 5 is a box diagram of the DE differences between seizure interval and seizure stage. Figure 6 is a box diagram of the HFD differences between seizure interval and seizure stage. It can be seen that the DE value in the seizure stage is lower than that in the seizure interval, and the value of HFD in seizure stage was higher than that in the seizure interval. This means that EEG sequence complexity is low during epileptic seizures.

In order to further evaluate the experimental performance of the method, sensitivity, specificity and area under the curve (AUC) were introduced as evaluation indexes. One crucial aspect of evaluating the performance of a classifier is to analyze its AUC value. The AUC is a valuable metric in assessing the classifier’s ability to discriminate between different classes. The higher AUC values signify that the KNN classifier is better at distinguishing between different classes or categories within a dataset [27]. 

Equation (20) is the sensitivity expression:(20)$$Sensitivity=\frac{TP}{TP+FN}$$

Equation (21) is the expression of specificity:(21)$$Specificity=\frac{TN}{TN+FP}$$
where TP represents the number of correctly classified positive samples, TN represents the number of correctly classified negative samples, FP represents the number of negative samples incorrectly classified and FN represents the number of positive samples incorrectly classified.

When DE and HFD were used as the feature input, 98.3% of the classification accuracy, 98.5% of the specificity, 98.9% of the sensitivity and 0.991 of the AUC value were achieved. This shows that the classification method has a good classification effect. The comparative experimental data are shown in Table 2. By comparing the results, it is found that DE and HFD as the feature group can improve the classification recognition performance.

The accuracy is compared with the classification task of full channels (23 channels) and the classification task of 5 channels obtained using only the GWO algorithm. The results of the comparison are shown in Table 3. By observing Table 3, it can be seen that the effect of selecting channels using only the GWO algorithm is similar to the classification accuracy of the full channel. This indicates that the algorithm in this paper works better, and the effectiveness of the algorithm in this paper is verified.

The method proposed in this paper is compared with Method A [22], Method B [23], Method C [24] and Method D [25], and a comparison of the four methods was conducted. Then, Table 4 and Figure 7 show the experimental results of the CHB-MIT database. The method proposed in this paper achieves the highest accuracy of 98.3% and the highest sensitivity of 98.9%. The experimental data show that the method proposed in this article achieves the best performance in all measurement indicators.

## 5. Conclusions

In this paper, variational modal decomposition and an improved grey wolf algorithm were used to detect epileptic EEG signals. The VMD method can reduce the nonstationarity of time series with high complexity and strong nonlinearity, so as to obtain relatively stationary subseries with multiple different frequency scales. Meanwhile, this method overcomes the problems of endpoint effects and modal component aliasing in the EMD method. The grey wolf optimization algorithm can be used to solve continuous optimization problems. The experimental data were extracted from each patient’s preseizure and seizure periods of 200 s, with every 2 s as a segment, meaning 100 data points were obtained for each patient’s health period as well as 100 data points for each patient’s epilepsy period. The results show that the proposed method has the highest accuracy (98.3%), the highest sensitivity (98.9%) and the highest specificity. The public dataset of CHB-MIT was selected for verification, and the final classification accuracy reached 98.3%. In order to make the algorithm performance fully verified, 10-fold cross-validation was used in the experiments. This verified that the results are consistent.

This study, as an automatic seizure detection approach, achieved effective classification results. It also highlighted significant differences in EEG signal complexity between the pre-seizure stage and the actual seizure stage. During epileptic seizures, the DE values decreased and HFD values increased, indicating changes in brain activity complexity. This difference may manifest as cognitive impairment in patients with epilepsy. The algorithm proposed in this paper can be used as an index to detect epileptic seizures and has certain guiding significance for the early diagnosis and effective treatment of epileptic patients. The method presented in this paper has high accuracy and sensitivity in the detection of epileptic EEG signals. While the proposed algorithm has yielded promising results in the classification of EEG signals, there are still limitations in visual representation, particularly in showcasing raw signals and artifact removal. In our forthcoming research endeavors, we are committed to procuring additional data resources and enhancing the performance of our method in visual presentation. In addition, collaboration with medical experts for clinical trials and real-time assay optimization is also on our research agenda, with the aim of improving the effectiveness of epilepsy diagnosis and treatment.

## Figures and Tables

**Figure 1 sensors-23-08078-f001:**
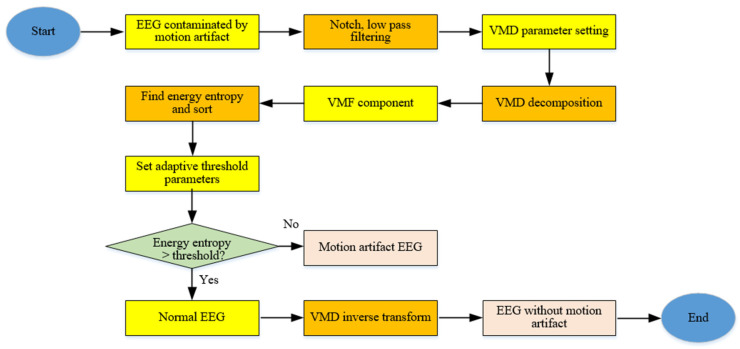
Flow chart of the VMD-VMF method.

**Figure 2 sensors-23-08078-f002:**
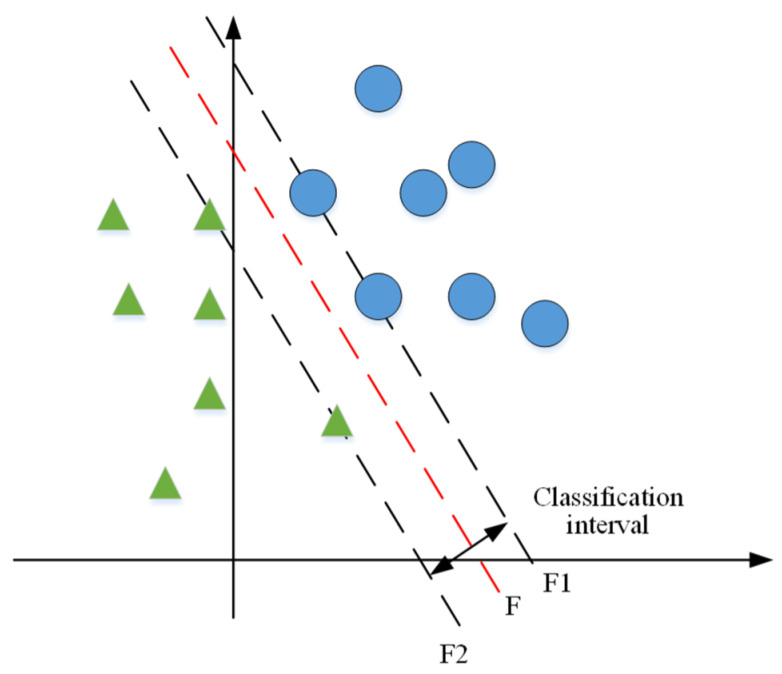
Optimal classification surface of the support vector machine. (The red line represents the middle line, which is the dividing line of different types. The blue circles represent a normal EEG sample of a healthy person in a conscious state. The green triangles represent the EEG samples of epilepsy patients at the onset of the disease).

**Figure 3 sensors-23-08078-f003:**
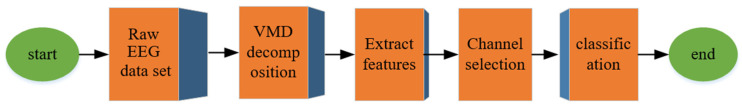
Classification and detection flow chart of the EEG signals.

**Figure 4 sensors-23-08078-f004:**
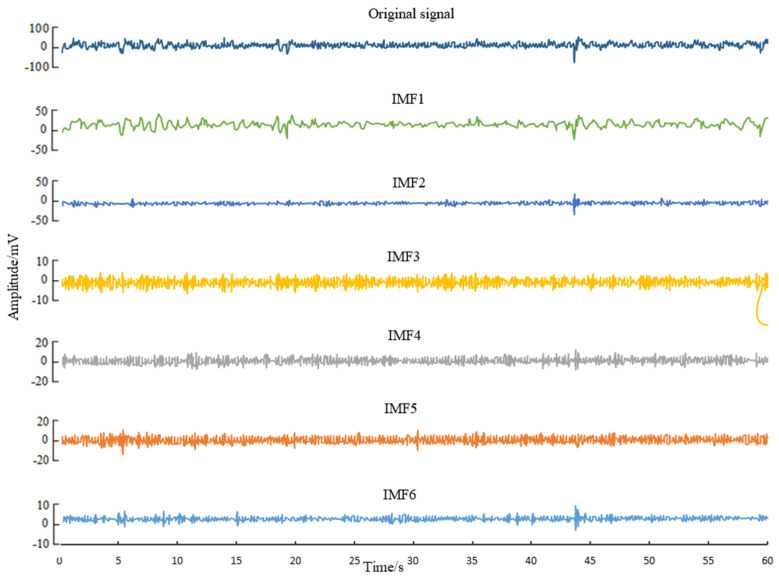
The VMD decomposition results of the original EEG signal.

**Figure 5 sensors-23-08078-f005:**
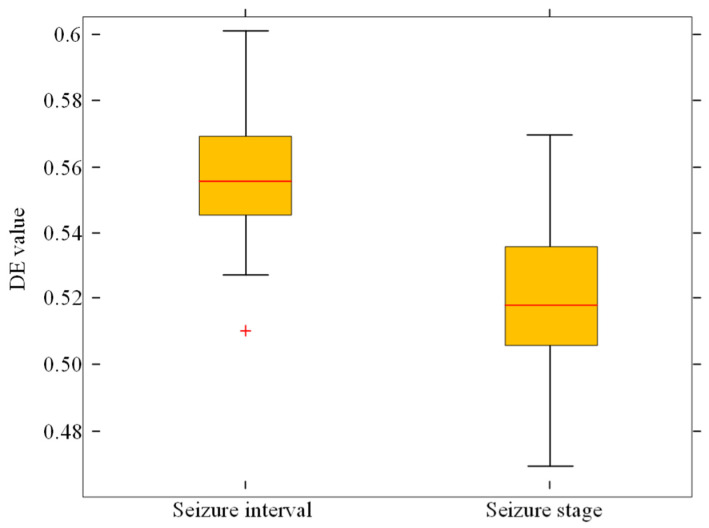
The DE value difference between seizure interval and seizure stage. (The “+” represents the location of the outlier).

**Figure 6 sensors-23-08078-f006:**
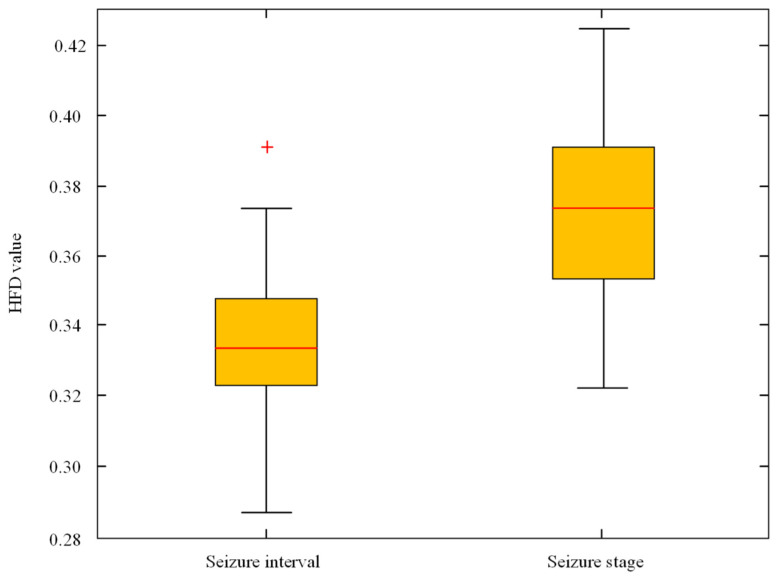
The HFD value difference between seizure interval and seizure stage. (The “+” represents the location of the outlier).

**Figure 7 sensors-23-08078-f007:**
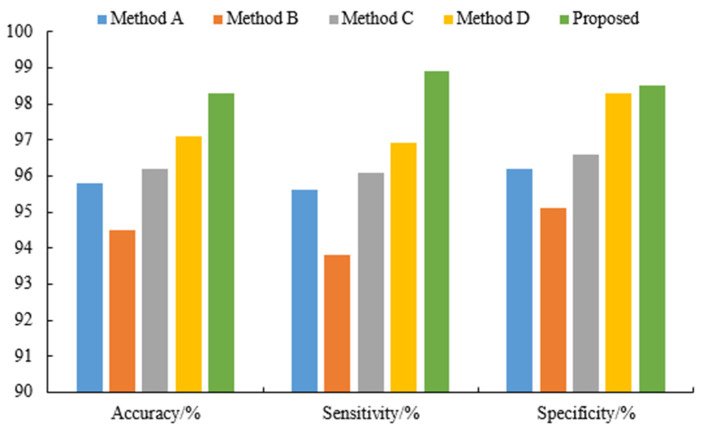
Comparison between the proposed method and the other five methods.

**Table 1 sensors-23-08078-t001:** Common epilepsy datasets.

Data	Number of Subjects	Total Number of Attacks	Signal Types	Sampling Frequency (Hz)	Total Time (h)
Freiburg	21	87	Intracranial EEG	256	708
CHB-MIT	22	163	The scalp EEG	256	844
Bonn	10	100	Intracranial EEG	256	708
Kaggle	2 (people)	48	Intracranial EEG	400	627
	5 (dog)			5000	
Barcelona	5	3750	Intracranial EEG	512	83

**Table 2 sensors-23-08078-t002:** Classification results under different features.

Indicators	DE	HFD	DE + HFD (Proposed)
Accuracy/%	97.9	98.1	98.3
Sensitivity/%	98.3	98.4	98.9
Specificity/%	98.5	98.4	98.5
AUC	0.987	0.988	0.991

**Table 3 sensors-23-08078-t003:** Classification results under different channels.

Indicators	Full Channels	Five Channels Selected by GWO (Proposed)
Accuracy/%	98.4	98.3
Sensitivity/%	99.0	98.9
Specificity/%	98.6	98.5
AUC	0.993	0.991

**Table 4 sensors-23-08078-t004:** Classification results under different features.

Methods	Accuracy/%	Sensitivity/%	Specificity/%
A	95.8	95.6	96.2
B	94.5	93.8	95.1
C	96.2	96.1	96.6
D	97.1	96.9	98.3
Proposed	98.3	98.9	98.5

## Data Availability

The labeled dataset used to support the findings of this study is available from the corresponding author upon request.

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
