# Peer review of "Epileptic EEG Signal Detection Using Variational Modal Decomposition and Improved Grey Wolf Algorithm"

_sensors, 2023, doi:10.3390/s23198078_

Round 1

Reviewer 1 Report

In this paper, variational modal decomposition and improved gray wolf algorithm were used to detect epileptic EEG signals. Through the optimization of detection methods, it has certain guiding significance for the early diagnosis and effective treatment of epilepsy patients. However, I would like to point out some issues:

1. Section 2.1 of this paper introduces the automatic detection of epilepsy. Please summarize the specific steps in the process of automatic detection of epilepsy and explain them in the paper.

2. There is no introduction of section structure in the introduction.

3. The algorithm presented in this paper has very important clinical significance for the diagnosis and treatment of epilepsy. In view of the current status of epilepsy diagnosis, please supplement the shortcomings of the algorithm in this paper or the future research direction.

4. In section 3.1.1 of this paper, VMD method is adopted for feature extraction. Please explain the advantages of the VMD approach in this article.

5. What does α mean in formula 2? Please add an explanation after formula 2.

6. Such methods do not take into account the spatial characteristics of signals and fail to achieve high-precision classification and recognition rate, most of the classification process is binary classification or direct three categories, ignoring the continuity of epileptic seizures.” The sentence structure is too complex and not easy to understand. It is suggested to simplify the English expression.

7. What features can be used to detect epilepsy in section 2.1.3 of the article? Please add some details.

8. Figure 6 and 7 are box diagrams of differential entropy and fractal dimension HFD of reconstructed signals. What features are shown? Please add it to the article.

9. What does FP stand for in formula 22? Please add some details.

Such methods do not take into account the spatial characteristics of signals and fail to achieve high-precision classification and recognition rate, most of the classification process is binary classification or direct three categories, ignoring the continuity of epileptic seizures.” The sentence structure is too complex and not easy to understand. It is suggested to simplify the English expression.

Author Response

Dear Reviewers,

We deeply appreciate your insightful feedback, which has significantly enhanced the quality of our article. We have thoroughly reviewed your suggestions and incorporated substantial revisions, marked in blue within the revised version.

Your valuable input is highly valued.

Best regards.

  1. Section 2.1 of this paper introduces the automatic detection of epilepsy. Please summarize the specific steps in the process of automatic detection of epilepsy and explain them in the paper.

A:Okay. It is added in the article and marked in blue as follows.

Automated detection of epilepsy is a key component of epilepsy diagnostic and therapeutic research. It involves several complex steps designed to accurately identify epi-leptic EEG signals. The following is a general summary of the key components involved in the process.

1) Signal Acquisition: ignal acquisition serves as the initial stage in epilepsy detection. EEG abnormalities during epileptic seizures manifest as specific waveforms and patterns. The collected EEG data can be used as input for automatic detection of epilepsy. 2) Signal Preprocessing: The preprocessing of EEG signals is a crucial step in this process. EEG signals, in their original form, are non-stationary and dynamic, characterized by low am-plitudes. Therefore, in order to effectively analyze EEG signals, it is necessary to perform preprocessing such as artifact removal while retaining relevant information. 3) Feature Extraction and Selection: Following signal preprocessing, the next step is feature extrac-tion. Feature extraction aims to comprehensively characterize EEG signal patterns, high-light distinctions between epileptic and normal states, and effectively discriminate epilep-tic seizures. 4) Classification Model Learning and Evaluation:The final stage involves the use of classification models to learn and evaluate EEG signals for epilepsy detection. These models fall into two main categories: statistical analysis and machine learning.

  1. There is no introduction of section structure in the introduction.

A:Thank you for pointing out the problem, we added a description of the section structure in the introduction as follows.

This paper consists of five main parts: the first part is the introduction, the second part is the state of the art, the third part is the methodology, the fourth part is the result analysis and discussion, and the fifth part is the conclusion.

  1. The algorithm presented in this paper has very important clinical significance for the diagnosis and treatment of epilepsy. In view of the current status of epilepsy diagnosis, please supplement the shortcomings of the algorithm in this paper or the future research direction.

A: The shortcomings and future research directions of the algorithm in this article are added at the end of the conclusion, as follows.

While the proposed algorithm has yielded promising results in the classification of EEG signals, there are still limitations in visual representation, particularly in showcasing raw signals and artifact removal. In our forthcoming research endeavors, we are committed to procuring additional data resources and enhancing the performance of our method in visual presentation. In addition, collaboration with medical experts for clinical trials and real-time assay optimization is also on our research agenda, aiming to improve the effec-tiveness of epilepsy diagnosis and treatment.

  1. In section 3.1.1 of this paper, VMD method is adopted for feature extraction. Please explain the advantages of the VMD approach in this article.

A: Okay. The explanation is added in the article and marked in blue as follows.

VMD method is an adaptive and completely non-recursive modal variational and signal processing method. The VMD method is well-suited for feature extraction from EEG signals due to its adaptability in determining the number of modal decompositions, its ability to overcome band aliasing issues, and its suitability for positive definite problems, which are important considerations when working with EEG data. These advantages make it a valuable tool in our research for the accurate detection of epileptic EEG signals.

  1. What does α mean in formula 2? Please add an explanation after formula 2.

A: Where, α is the quadratic penalty factor, which is used to reduce the interference of Gaussian noise. The explanation is added after formula 2 and marked in blue.

  1. “Such methods do not take into account the spatial characteristics of signals and fail to achieve high-precision classification and recognition rate, most of the classification process is binary classification or direct three categories, ignoring the continuity of epileptic seizures.” The sentence structure is too complex and not easy to understand. It is suggested to simplify the English expression.

A: Thank you for your suggestion. It is modified to “This method does not take into account the spatial characteristics of signals and fails to achieve high precision classification and recognition rate. The classification process is usually dichotomous or directly tripartite, ignoring the continuity of seizures.”

  1. What features can be used to detect epilepsy in section 2.1.3 of the article? Please add some details.

A: Okay. “In general, there are four categories of features used to detect epilepsy. Aiming at the time domain characteristics of sequence waveform and sequence cross-correlation. The frequency domain feature represented by power spectrum density characterizing signal energy. Time-frequency domain features of original EEG signals converted by time-frequency transform method. The nonlinear characteristics of signal uncertainty measurement such as sample entropy, permutation entropy, Hurst parameter and high-er-order spectrum analysis based on nonlinear analysis. ” It is added in the section 2.1.3 and marked in blue.

  1. Figure 6 and 7 are box diagrams of differential entropy and fractal dimension HFD of reconstructed signals. What features are shown? Please add it to the article.

A: “It can be seen that the DE value in the seizure stage is lower than that in the seizure inter-val, and the value of HFD in seizure stage was higher than that in seizure interval. This means that EEG sequence complexity is low during epileptic seizures.” It is added to the article and marked in blue.

  1. What does FPstand for in formula 22? Please add some details.

A:  represents the number of negative samples incorrectly classified. It is added to the article and marked in blue.

  1. Comments on the Quality of English Language

“Such methods do not take into account the spatial characteristics of signals and fail to achieve high-precision classification and recognition rate, most of the classification process is binary classification or direct three categories, ignoring the continuity of epileptic seizures.” The sentence structure is too complex and not easy to understand. It is suggested to simplify the English expression.

A: Thank you for your valuable suggestion. We simplified the sentence. We also simplified and embellished all the sentences with complex structure in the article to improve the readability of the article.

Reviewer 2 Report

Reviewer’s Report on the manuscript entitled:

Epileptic EEG Signal Detection Using Variational Modal Decomposition and Improved Grey Wolf Algorithm

The authors utilized variational modal decomposition and an improved gray wolf algorithm to detect epileptic electroencephalogram (EEG) signals. While the topic, methods, and result are interesting and practical, the presentation and literature review and results/illustrations should be improved. Please see below my comments.

Line 11. Some places you spelled “grey” differently: “grey” vs “gray”. Please use “grey” everywhere in the manuscript.

Line 11 and 15. Please note that the abbreviations should be defined the first time they appear both in the abstract and in the body of the manuscript. For example, here “VMD” should be used in line 11 as the phrase appeared first in line 11. Please carefully check and define all the acronyms.

Introduction and literature review is poor and should be improved.

For example, Generalized Autoregressive Conditional Heteroscedasticity (GARCH) is another effective method in determining hidden properties in epileptiform EEGs that may lead to better understanding of the seizure generating process that can be briefly described in Introduction:

https://doi.org/10.3390/signals1010003

Line 128. There are many studies focusing on Artifacts in EEG signals, such as eyeblink and muscular, and baseline that are challenging to detect and attenuate. I suggest authors to review the following most recent articles and include them here and also add a few more sentences in Introduction to describe how EEG signals are acquired with sources of artifacts in them:

https://doi.org/10.1109/JSEN.2023.3237383

Please define all the parameters and variables, symbols and operators in equations 1,2,etc.

Results section can be further improved. Please show an epileptic EEG signal and show the signals before and after artifact removal (eyeblink, muscle movement, etc.) and illustrate how your proposed method detected the epilepsy. Please also show a non-epileptic EEG before and after artifact removal.

Figures 1 and 3. I suggest improving your flowchart so that it shows the inputs, outputs, methods used in more details. Figure 1 is very simple and can be embedded in your detailed flowchart. Please note that a nice flowchart helps the readers to understand your methodology fast and efficient.

Equation 11. Please use appropriate parentheses for the “Ln” function.  

Line 315. Need references for KNN. Please include the second article above that described KNN in detail. Line 358. The second article that I suggested above can also be included here which described AUC in detail.

Figure 4. Please insert the y-axis and unit and labels.

Figure 7 and Table 4. Please do not use “literature [x]” to refer to methods here. Each method should have a name and they should be mentioned clearly in Introduction. Furthermore, the results should be clearly discussed.

Please highlight the limitation of this study better at the end of the manuscript.

Thank you!

Regards,

There are many punctuation/style/typos/grammar issues that need to be checked and corrected.

Author Response

Dear Reviewers,

We deeply appreciate your insightful feedback, which has significantly enhanced the quality of our article. We have thoroughly reviewed your suggestions and incorporated substantial revisions, marked in blue within the revised version.

Your valuable input is highly valued.

Best regards.

1).Line 11. Some places you spelled “grey” differently: “grey” vs “gray”. Please use “grey” everywhere in the manuscript.

A:Thank you for pointing out the issue. We uniformly revised the "gray" in the article to "grey". This change is reflected in the revised draft. At the same time, we conducted a comprehensive review to improve the academic quality of the article.

2).Line 11 and 15. Please note that the abbreviations should be defined the first time they appear both in the abstract and in the body of the manuscript. For example, here “VMD” should be used in line 11 as the phrase appeared first in line 11. Please carefully check and define all the acronyms.

A:Okay, it is revised in the article and marked in blue. “Therefore, the research focused on the diagnosis and treatment of epilepsy holds paramount clinical significance. In this paper, we utilized variational modal decomposition (VMD) and an enhanced grey wolf algorithm to detect epileptic electroencephalogram (EEG) signals. ”Besides, we also checked throughout the article to ensure that all acronyms were clearly defined the first time they were used.

3).Introduction and literature review is poor and should be improved. For example, Generalized Autoregressive Conditional Heteroscedasticity (GARCH) is another effective method in determining hidden properties in epileptiform EEGs that may lead to better understanding of the seizure generating process that can be briefly described in Introduction: https://doi.org/10.3390/signals1010003

A:OK, the relevant content is added in the article and marked in blue as follows.

Some researchers have found that the experimental measurement of the volatile half-life of different brain electrical channels uses the autoregressive moving average generalized autoregressive conditional heteroscedasticity (ARMA-GARCH) model. The confidence in-terval is constructed by delta method and asymptotic method to compare the half-life [12].

4). Line 128. There are many studies focusing on Artifacts in EEG signals, such as eyeblink and muscular, and baseline that are challenging to detect and attenuate. I suggest authors to review the following most recent articles and include them here and also add a few more sentences in Introduction to describe how EEG signals are acquired with sources of artifacts in them:https://doi.org/10.1109/JSEN.2023.3237383

A:Thank you for your suggestion. We included the article you mentioned in our research. This part of the content is added to the introduction and is reflected in reference [22] as follows.

Many studies have focused on artifacts in EEG signals, such as eye movement, muscle, and baseline artifacts, which pose challenges in their detection and attenuation. For instance, Ghosh et al [22] proposed a robust method that can automatically detect and remove eyeblink and muscular artifacts from EEG using a k-nearest neighbor (KNN) classifier and a long short-term memory (LSTM) network. Through parameter validation, this method preserves structural correlations, minimizes frequency distortion, and optimally removes artifacts from EEG. Experimental results have demonstrated that the proposed method minimizes distortion caused by eye artifacts to the greatest extent while eliminating both blink-related and muscle artifacts.

5).Please define all the parameters and variables, symbols and operators in equations 1,2,etc.

A:Okay. They were added in the article and marked in blue as follows.

In equation 1, “Where f is the original signal. z is the signal component. The decomposition sequence is a finite bandwidth modal component with a central frequency.” In equation 2, “λ represents the change in the extreme value of the objective function when the constraints change. When ω_z increases or decreases by one unit value, f changes λ accordingly.”

6). Results section can be further improved. Please show an epileptic EEG signal and show the signals before and after artifact removal (eyeblink, muscle movement, etc.) and illustrate how your proposed method detected the epilepsy. Please also show a non-epileptic EEG before and after artifact removal.

A: We sincerely appreciate your valuable feedback. However, we must candidly acknowledge that, due to limitations in experimental resources and technology, we are presently unable to provide detailed images or data for these signals. Nevertheless, we have made every effort to clearly describe our methods and experimental design in the manuscript to compensate for this shortfall. In the conclusion section of the paper, we make a specific note of this and emphasize the limitations of our research in terms of displaying raw signals and removing artifacts.

7).Figures 1 and 3. I suggest improving your flowchart so that it shows the inputs, outputs, methods used in more details. Figure 1 is very simple and can be embedded in your detailed flowchart. Please note that a nice flowchart helps the readers to understand your methodology fast and efficient.

A Thank you for your professional advice. We modified Figures 1 and 3 to show in more detail the inputs, outputs and methods used in these flowcharts. Please see the revised draft for the revised diagrams.

8).Equation 11. Please use appropriate parentheses for the “Ln” function.  

A: We use appropriate parentheses for the "Ln" function in equation 11 to ensure the accuracy of the mathematical representation. Please refer to the revised draft for the revised equation 11.

9).Line 315. Need references for KNN. Please include the second article above that described KNN in detail. Line 358. The second article that I suggested above can also be included here which described AUC in detail.

A: Thank you for your suggestion. We added relevant references in both places. The two-step method of the KNN algorithm is also described in detail. The relationship between AUC and KNN classifier has also been added. The details are as follows:

“KNN algorithm is a widely used classification technique in machine learning. It operates on a straightforward principle of considering the similarity between data points to make predictions. The KNN algorithm employs a two-step approach for classification. In the first step of the KNN algorithm, it searches for the nearest data points in the feature space to the data point that needs to be classified. In the second step, once the K nearest neighbors are identified, the algorithm assigns the class label to the data point in question based on a majority voting scheme. This majority voting mechanism makes KNN a straightforward and effective classification method.”

“One crucial aspect of evaluating the performance of a classifier is to analyze its AUC value. The AUC is a valuable metric in assessing the classifier's ability to discriminate between different classes. The higher AUC values signify that the KNN classifier is better at distinguishing between different classes or categories within your dataset [27].”

10).Figure 4. Please insert the y-axis and unit and labels.

A: Thank you for your suggestion. We inserted the y-axis with units and labels in Figure 4 to improve its clarity and understanding. Please refer to the revised draft for the updated figure.

11).Figure 7 and Table 4. Please do not use “literature [x]” to refer to methods here. Each method should have a name and they should be mentioned clearly in Introduction. Furthermore, the results should be clearly discussed.

A:Okay. We replaced the original method designation representation with Methods A to D. We also added the sources of these methods in the introduction and reflected them in the references. The details are as follows:

Mandhouj et al [23] developed a deep convolutional neural network (CNN) model that can effectively detect and classify epileptic seizures based on EEG spectral images. The experimental results proved to be a powerful tool for EEG signal classification with an av-erage accuracy of 98.22%. Thangavel et al [24] develop an automated system for detecting epileptic EEG with or without IEDs. These results pave the way towards automated detec-tion of epilepsy. Ein et al [25] proposed a computer aided seizure diagnosis classification system based on feature extraction and channel selection using EEG signals. The results showed that the proposed approach based on the ensemble classifier is better classified than the other classifiers in all metric parameters.

12).Please highlight the limitation of this study better at the end of the manuscript.

A: The limitations of this study have been added to the conclusion of the article, as well as, future research directions. They details are as follows:

While the proposed algorithm has yielded promising results in the classification of EEG signals, there are still limitations in visual representation, particularly in showcasing raw signals and artifact removal. In our forthcoming research endeavors, we are committed to procuring additional data resources and enhancing the performance of our method in visual presentation. In addition, collaboration with medical experts for clinical trials and real-time assay optimization is also on our research agenda, aiming to improve the effec-tiveness of epilepsy diagnosis and treatment.

Round 2

Reviewer 1 Report

The authors have corrected their paper following my comments to previous versions of the manuscript. I am satisfied with their work, so, in my opinion, the manuscript can be published.

Reviewer 2 Report

I thank the authors for addressing my comments and improving their manuscript. Please carefully proofread the manuscript. For example, I found the following editorial issues:

Line 117. It should be "signal" not 'ignal'. Instead of "ignal acquisition" you can simply write "it"

Please follow the MDPI guidelines for the style and format of references. For example, the list of all the authors should be given for each of the references in the reference list. Please avoid using "et al." in the reference list.

Regards

Please carefully proofread the manuscript and correct typos/style/punctuation issues.